# Impact of Dietary Fat on the Progression of Liver Fibrosis: Lessons from Animal and Cell Studies

**DOI:** 10.3390/ijms221910303

**Published:** 2021-09-24

**Authors:** Fangping Jia, Xiao Hu, Takefumi Kimura, Naoki Tanaka

**Affiliations:** 1Department of Metabolic Regulation, Shinshu University School of Medicine, Matsumoto 390-8621, Japan; 17mh204b@shinshu-u.ac.jp; 2Department of Pathophysiology, Hebei Medical University, Shijiazhuang 050017, China; 19001442@hebmu.edu.cn; 3Department of Gastroenterology, Shinshu University School of Medicine, Matsumoto 390-8621, Japan; kimuratakefumii@yahoo.co.jp; 4International Relations Office, Shinshu University School of Medicine, Matsumoto 390-8621, Japan; 5Research Center for Social Systems, Shinshu University, Matsumoto 390-8621, Japan

**Keywords:** fatty liver, NASH, dietary fat, cholesterol, saturated fatty acid, trans fatty acid

## Abstract

Previous studies have revealed that a high-fat diet is one of the key contributors to the progression of liver fibrosis, and increasing studies are devoted to analyzing the different influences of diverse fat sources on the progression of non-alcoholic steatohepatitis. When we treated three types of isocaloric diets that are rich in cholesterol, saturated fatty acid (SFA) and trans fatty acid (TFA) with hepatitis C virus core gene transgenic mice that spontaneously developed hepatic steatosis without apparent fibrosis, TFA and cholesterol-rich diet, but not SFA-rich diet, displayed distinct hepatic fibrosis. This review summarizes the recent advances in animal and cell studies regarding the effects of these three types of fat on liver fibrogenesis.

## 1. Introduction

Non-alcoholic fatty liver disease (NAFLD) is a disease entity that includes non-alcoholic fatty liver (NAFL, presence of hepatic steatosis without other liver damage) and non-alcoholic steatohepatitis (NASH, infiltration of inflammatory cells, presence of hepatocellular degeneration with and without irreversible fibrosis), and NAFLD may progress into liver cirrhosis and cancer [1,2,3,4,5]. The most widespread hypothesis, by which NAFL progresses to NASH, is the “multiple-hit” hypothesis. The “first hit” is the fat accumulation in hepatocytes. The steatotic hepatocytes become more vulnerable due to the “first hit”. Lipid peroxidation, oxidative stress, pro-inflammation cytokine and mitochondrial dysfunction are the “second multiple hits”, which promote the inflammation, oxidative stress, endoplasmic reticulum (ER) stress, and fibrosis [6,7].

Several epidemiological studies have demonstrated that dietary factors can affect the clinical course of liver disease [8,9,10]. For example, in 9221 patients with chronic hepatitis C during 13.3 years of follow-up, participants who reported a diet high in protein had a significantly higher risk of hospitalization or death due to cirrhosis or liver cancer after adjusting for potential confounders [11]. Although total fat consumption was not remarkably associated with the risk of cirrhosis or liver cancer, a significant relationship was detected for cholesterol consumption [12]. However, due to the complexity of lifestyle and dietary habits, it remains difficult to clarify which dietary habits contribute to a diminished outcome and the precise mechanisms involved. Especially, liver fibrogenesis and tumorigenesis are progressive in the long term, with complicated mechanisms. Therefore, animal and cell studies would provide clue to clarify the impact of specific dietary fat on the progression of liver disease [13]. 

To uncover the influence of a diet enriched in cholesterol, saturated fatty acids (SFA), and trans fatty acids (TFA) on hepatic steatosis and ensuing liver disorders, hepatitis C virus core gene transgenic (HCVcpTg) mice that spontaneously developed hepatic steatosis without apparent fibrosis were used [14,15,16,17]. When we treated three types of isocaloric diets that are rich in cholesterol, SFA, and TFA with HCVcpTg mice, TFA and cholesterol-rich diet, but not SFA-rich diet, displayed the distinct hepatic fibrosis [18,19,20]. Such diets are closely associated with cardiovascular disease, insulin resistance, diabetes mellitus, colorectal cancer, and metabolic syndrome [21,22]. Wang et al. demonstrated that the hazard ratios of total mortality among specific dietary fats were the highest in TFA and SFA, and replacing 5% of energy from SFA with equivalent energy from unsaturated fats was estimated to reduce total mortality after adjustment for known and suspected risk factors [23]. These findings support the notion that different types of dietary fats have various impacts on total- and cause-specific mortality, including cancer, cardiovascular disease, and end-stage liver diseases. Liver is a main organ to metabolize dietary fats, and we have investigated the impact of dietary lipids, especially cholesterol, SFA, and TFA. Therefore, in this review, we focused on these three lipids and summarized their influence on liver fibrosis and the possible mechanisms.

## 2. Role of Cholesterol in the Progression of Liver Fibrosis

Cholesterol is a highly toxic lipid because of the potential to form crystals in the cytoplasm [24,25]. Therefore, increased intracellular cholesterol levels influence various types of cells, not only hepatocytes, but also Kupffer cells (KC), hepatic stellate cells (HSC), cholangiocytes, and liver sinusoidal endothelial cells (LSECs). These cells interact with each other and play important roles on the progression of liver fibrosis (Figure 1). 

### 2.1. Cholesterol’s Action for Hepatocytes

Free cholesterol accumulation has toxic effects on hepatocytes [26,27,28], promoting progression of fibrosis through multiple mechanisms; thus, cholesterol is esterified with fatty acids. A high-cholesterol diet induces cholesterol crystals form within large lipid droplets (LDs) in hepatocyte. The crystallization activates NLR family pyrin domain containing 3 inflammasome, which contributes to the secretion of chemokines and inflammatory cytokines, such as interleukin (IL)-1β. The remnant LDs of pyroptotic necrotic hepatocytes are processed by “crown-like structures” formed with KCs, and the KCs are transformed into activated lipid-laden foam cells. The foam cells activate stellate cells by chemotactic and inflammation signals, leading to fibrogenesis [24,25]. The recent research observed that simple steatosis did not exhibit cholesterol crystals, and the cholesterol crystallization and “crown-like structures” formed by KCs promoted the progression of steatosis to NASH [29]. 

Feeding high-cholesterol diet to rats raised a generation of reactive oxygen species (ROS) due to the increased p47^PHOX^ and p22^PHOX^ subunits of nicotinamide adenine dinucleotide phosphate oxidase; meanwhile, the mRNA expression of antioxidant enzymes was reduced [30]. However, in the mouse model, despite the enhanced oxidative stress was observed by cholesterol overload, the mRNA levels of ROS-generating and ameliorating enzymes were increased simultaneously [18]. A growing number of studies proved that dietary high cholesterol enhanced free cholesterol accumulation in ER, which triggered ER stress in hepatocytes [31,32]. In addition, the association between enhanced apoptosis and cholesterol overload was proven by in vivo and in vitro studies [18,33,34]. The dysregulated cellular stresses and promoted apoptosis are correlative with aggravated hepatocyte damage and fibrogenesis. 

### 2.2. Cholesterol’s Action for KC

KCs are the resident macrophages of liver and represent 20–25% of all liver cells [35]. Previous studies demonstrated that lysosomal cholesterol accumulation in KCs is closely correlated with hepatic inflammation and fibrogenesis [8,36,37]. KCs uptake nonmodified cholesterol-rich low-density lipoprotein (LDL) through binding its receptor (LDLR), and then combines with lysosomes in the form of endosomes. By lysosomal enzymes, the cholesterol ester is hydrolyzed into free cholesterol, which can be transferred into cytoplasm. The elevated concentration of cellular cholesterol inhibits the expression of *Ldlr* gene, which can suppress the activation of KCs [38,39,40]. 

The mechanism underlying uptake of cholesterol-rich LDL, such as oxidized LDL (oxLDL), is different from nonmodified LDL. The oxidized LDL is internalized by scavenger receptors: cluster differentiation 36 (CD36, fatty acid translocase) and scavenger receptor A (SR-A) [41,42]. The concentration of cellular oxidized LDL does not affect the expression of CD36 and SR-A, which contribute to the continuous uptake of oxLDL and formation of foam cells. 

To investigate the pro-inflammatory and pro-fibrogenic effects of oxLDL, the high-fat, high-cholesterol diet-fed LDL receptor-null (*Ldlr*^−/−^) mice were immunized with anti-oxLDL immunoglobulin M antibodies [43,44,45]. The immunized mice exhibited a decreased number of infiltrated macrophages, neutrophils, and T-cells; meanwhile, the inflammatory markers: tumor necrosis factor (TNF), IL-6, and monocyte chemoattractant protein, were significantly reduced in the liver. In addition to inhibited inflammation, the immunized *Ldlr*^−/−^ mice exhibited the lower fibrosis-related gene expressions, such as collagen type 1A1 (COL1A1) and transforming growth factor (TGF) β [43]. Consistently, the HCVcpTg mice fed with 1.5% cholesterol diet revealed the increases in KCs and more severe inflammation [18]. These data uncovered that cholesterol overload plays a role on KCs activation and the progression from NAFL to NASH with severe fibrosis.

### 2.3. Cholesterol’s Action for HSC

HSCs are the major effector cells during hepatic fibrogenesis [46,47]. The high levels of lectin-like oxidized LDL receptor-1 (LOX-1) and oxLDL can be observed in hypercholesterolemic patients [48,49,50,51]. Previous studies have proved that oxLDL taken by LOX-1 plays a role on HSC activation [52]. Additionally, the intracellular free cholesterol overload enhances the HSC activation, contributing to the increased Toll-like receptor (TLR) 4 levels through the suppressed endosomal–lysosomal degradation pathway of TLR4. The elevated TLR4 levels sensitized HSCs to TGFβ [53,54]. Cholesterol acyltransferase (ACAT) is the enzyme that catalyzes the conversion of free cholesterol to cholesterol ester. Studies uncovered that hepatic fibrosis is more severe in *Acat1*-deficient mice than wild-type mice; meanwhile, the hepatocellular damage, inflammation and KC activation were comparable between groups [55]. Except for the free cholesterol overload-induced HSC activation, the increased *Tnf* and *Tgfb* expression, due to the KCs activation, can activate HSCs directly [56,57,58]. These studies illustrate that oxLDL accumulation, free cholesterol overload, and the interaction of KCs significantly affect the HSC activation, and thus alter the fibrosis degree.

### 2.4. Cholesterol’s Action for Cholangiocytes

Ductular reaction (DR) is defined as the proliferation of cholangiocytes and hepatic progenitor cells, and studies have proved the DR is closely correlated with fibrosis [59,60,61]. The 1.5% cholesterol diet-fed HCVcpTg mice presented the more severe bile duct proliferation than AIN93G diet-fed group, as revealed by anti-cytokeratin 19 immunohistochemistry, which is often used to identify cholangiocytes [18]. There is no direct evidence about the influence of cholesterol on cholangiocytes. More models should be established to clarify the mechanism underlying enhancement of DR by high cholesterol diet. 

### 2.5. Cholesterol’s Action for LSECs

Healthy LSECs display the anti-inflammatory and anti-fibrogenic properties through maintaining the quiescence of KCs and HSCs. In NAFLD, LSECs display the dysfunction and capillarization, which can promote fibrosis [62,63]. Few studies uncovered the direct effects of free cholesterol accumulation on LSECs, while previous studies have demonstrated that a high cholesterol diet exacerbated by acetaminophen-induced acute liver injury in a TLR9/inflammasome-dependent manner [64]. Further studies are needed to evaluate the significance of free cholesterol accumulation on LSECs dysfunction and capillarization, thereby leading to the progression of fibrosis. 

## 3. Role of SFA in the Progression of Liver Fibrosis

### 3.1. SFA’s Action for Hepatocytes

In the progression of NAFLD, hepatic triglyceride (TG) accumulation is recognized as the first hit, which is the prerequisite for second hit, involving in oxidative and metabolic stress [1,2,6,7,9]. The increased SFA, but not unsaturated fatty acids, promoted the ER stress and apoptosis in hepatocytes (Figure 2). Increasing evidence proved that ER stress was of critical importance to SFA-induced cellular dysfunction and apoptosis [65,66]. Previous studies provided us with the evidence that SFA dysregulated mitochondrial metabolism and thereby elevated production of ROS. Of note, Jun-N-terminal kinase (JNK) activation was involved in apoptosis, which was related to oxidative stress and ER stress [67,68], and activation of JNK1, but not JNK2, was correlated with SFA-induced apoptosis [69]. In contrast, previous studies clarified that C/EBP-homologous protein, a pro-apoptosis maker, was not essential for SFA-induced apoptosis [65,70]. The action of HSCs and KCs on uptake of apoptotic cells enhances the expression of pro-fibrogenic genes. 

In the early stage of NAFLD, TG accumulation is the hallmark. However, TG, but not free fatty acid (FFA), is the protective form to prevent progression of fibrosis [71,72]. Diacylglycerol acyltransferase 2 (DGAT2) is an enzyme responsible for catalyzing the final step of TG synthesis in the liver. Yamaguchi et al. demonstrated that treatment of mice with DGAT2 antisense oligonucleotide attenuated the hepatic steatosis but worsened the fibrosis and liver damage [71]. Consistent with results above, Mei et al. revealed that palmitic acid (PA) induced lipoapoptosis and decreased autophagy, accelerating the progression of fibrosis [73]. Overall, these data uncovered the crucial role of SFA on fibrosis by dysregulated ER stress, mitochondrial metabolism, lipoapoptosis, and autophagy in hepatocytes. 

### 3.2. SFA’s Action for KC

Macrophage phenotypes are classified into two subtypes: M1 and M2. M1-polarized macrophage was induced by lipopolysaccharide and interferon γ, producing pro-inflammatory cytokines and chemokines, thereby promoting inflammation. Conversely, M2 possess the properties of inflammation depression and tissue repair through the stimulation by IL-4 and IL-13 [74,75]. Of note, several studies have revealed the anti-fibrogenic properties of M2 [76]. 

Luo et al. reported that SFA induced the M1 polarization, while n-3 polyunsaturated fatty acid (PUFA) polarized the KC into M2 in in vivo and in vitro experiments [77]. Besides lipid overload outside of KC, Leroux et al. revealed that overload inside of KC displayed pro-inflammatory phenotypes [78]. Peroxisome proliferator-activated receptor (PPAR) γ possesses the capacity of shifting the M1 to M2, and the activation of PPARγ may improve the SFA-induced M1 polarization and NASH [77,79,80,81]. Moreover, previous studies proved that IL-10 released by M2 promoted the apoptosis of M1 [82]. These data imply that interventions targeting M2 polarization provide us with an attractive strategy for suppression of NASH progression. Cyclooxygenase (COX2) is overexpressed in tumor tissues, including tumor cells and macrophages [83,84]. Lee et al. presented that SFA, but not unsaturated fatty acid, interacted with TLR4 as an endogenous ligand, and then induced the activation of COX2 due to upregulating nuclear factor-kappa B (NF-κB) [85]. Certain studies proved that COX2 was involved in liver fibrogenesis [86]. The above data prompted us to conclude that regulation of the M2/M1 balance is the potent method to improve NASH or liver fibrosis.

### 3.3. SFA’s Action for HSC

Increasing studies have investigated the role of hepatocellular SFA accumulation on HSC activation. Wobser et al. revealed that the conditioned media (CM) of steatotic hepatocytes incubated with palmitate initiated the activation of HSCs, as evidenced by the increased production of α-smooth muscle actin (α-SMA) and type 1 collagen [87]. Meanwhile, the upregulated expression of TGF-β, tissue inhibitor of metallo-proteinase-1 (TIMP-1), TIMP-2 and matrix metalloproteinase-2 were also observed in activated HSCs. Additionally, CM of steatotic hepatocytes promoted the HSC proliferation and inhibited apoptosis. The soluble mediators secreted by steatotic hepatocytes, but not residual palmitate, promoted hepatic fibrosis. Conversely, palmitate singularly decreased the HSC proliferation and expression of α-SMA and collagen [87]. Moreover, previous studies clarified that incubation of palmitate elevated the levels of extra and intracellular sphingosine 1-phosphate (S1P) in hepatocytes, thereby evoking the HSC activation through S1P3 receptor [88]. One more study proved succinate overload accelerated hepatic fibrosis through succinate/hypoxia-inducible transcription factor-1α (HIF-1α) axis, which can be suppressed by curcumin [89]. n-3 PUFA presented anti-fibrogenic effects in HSCs [90]. The precise mechanisms provide us with more insight into hepatic fibrosis.

### 3.4. SFA’s Action for Cholangiocytes

Recent studies focused on the correlation between SFA and biliary damage. Emerging evidence revealed that FFA induced cholangiocyte lipoapoptosis [91]. Exposure of cholangiocytes to FFAs promoted the expression of phosphorylated p38-mitogen activated protein kinase and extracellular signal-regulated kinase (ERK), with the subsequent nuclear localization of forkhead family transcription factors 3 (FOXO3). The nuclear FOXO3 enhanced the expression of p53-up-regulated modulator of apoptosis (PUMA), and the downstream caspase-3/7 triggered lipoapoptosis [91]. In addition to PUMA, FoxO3 increased miR-34a to cause palmitate-induced cholangiocyte lipoapoptosis, mainly through depressed sirtuin 1, receptor tyrosine kinase, and Kruppel-like factor 4 [92]. Obese Zucker rats exhibited defective hepatobiliary transport capacity, which may contribute to more susceptibility to liver injury [93]. These data provide us with therapeutic feasibility targeting amelioration of FFA toxicity in cholangiocytes. 

### 3.5. SFA’s Action for LSECs

Previous studies have clarified that the stimuli, such as TG, FFA, and ethanol, triggered endothelial dysregulation. The endothelial cells become capillarized and processes the pro-inflammatory and pro-fibrotic phenotype [62,63]. The damaged endothelial cell recruits natural killer T (NKT) cells, B cells and KCs. Alexander et al. have proved that NKT cell migration and functionality in liver fibrosis are controlled by chemokine (C-X-C motif) ligand 6 (CXCL6) and its receptor CXCR6, and liver fibrosis was improved in *Cxcr6*-deficient mice [94]. The present studies imply that B cells play an important role on liver, skin and lung fibrogenesis [95]. The previous studies have uncovered that in SECs, the TGFβ1 promoted the synthesis of connective tissue proteins and interstitial fibrillar matrix protein [96]. In vivo experiment, the SECs isolated from CXCL4-damaged livers presented the comparable results from in vitro studies, in which the SECs were stimulated by TGFβ1. Besides indirect pro-fibrotic function, the main contribution to fibrogenesis is HSC activation through the secretion of fibronectin EIIIA, TGFβ and platelet-derived growth factor [96]. For the abovementioned reasons, protecting the LSEC from various lipid-associated mediators and stress may become an attractive strategy to attenuate liver fibrosis. 

## 4. Role of TFA in the Progression of Fibrosis

Unsaturated fatty acids with at least one double bond in the trans configuration was designated as TFA. TFA is naturally present at low levels in dairy products and animal meat, while industry-generated TFA is contained in such hardened vegetable fats as margarine and shortening as well as in snack foods and fried foods. The World Health Organization estimated that excessive TFA consumption led to more than 500,000 deaths yearly from cardiovascular disease, calling on governments to promote the REPLACE project to eliminate dietary TFAs. Although excessive TFA intake is associated with cardiovascular disease, shorter life expectancy and cognitive disorders [97,98], the previous studies have established that intake of TFA causes many detrimental effects in hepatic diseases. 

To check the influence of dietary TFA on hepatic steatosis and ensuing hepatocarcinogenesis, male HCVcpTg mice were treated with an isocaloric TFA-rich diet that replaced most of the soybean oil with shortening for 5 months. The TFA-fed groups presented notably hepatic fibrosis. Furthermore, the hepatic expression of fibrosis-related factors, such as COL1A1, α-SMA, and connective tissue growth factor (CTGF), were also enhanced [20]. The expression of CTGF is highly upregulated in hepatofibrosis but not in normal liver as a main signal factor in the activation and proliferation of HSC [99]. COL1A1 and α-SMA are hallmarks of activated HSC, and principally induce the generation and deposition of the extracellular matrix [100]. 

Similar effects of TFA were also observed in other studies. In the ALIOS-diet (45% fat, of which 30% is TFA)-fed mice, intense fibrosis emerged from high levels of hepatic fibrosis percentage, and proliferation of collagen fibers were accompanied with high expression of fibrogenesis-related genes [101]. In adult male C57BL/6 mice, after treatment with a high calorie-diet that mainly contained medium-chain TFAs for 16 weeks, the distribution of hepatic fibrosis shown in sections were similar to biopsied findings of NASH patients, with the fibrosis appearing predominantly in zone 1 or the perisinusoidal space. Meanwhile, the expression of TGF-β1 and α-SMA was elevated [102], indicating that TGF-β1 signaling pathway may involve in TFA-associated fibrotic injury. Additionally, long-term feeding of high fructose in combination with TFA to rats induced fibrosis, as evidenced by high contents of hydroxyproline and increased expression of fibrogenic genes [103]. Notwithstanding, the abovementioned studies may not clarify that TFA straightly participated in liver fibrogenesis, they may demonstrate that intake of TFA promoted the development of hepatic fibrosis. 

Although there are several studies that can forcefully confirm that TFA plays a potential important role in development of liver fibrosis, there is limited proof on how TFA causes fibrogenesis in each type of liver cells. Therefore, in this section, we focused on the influence of TFA on drivers of liver fibrosis, such as inflammatory signaling and aberrant lipid metabolism (Figure 3).

### 4.1. The Impact of TFA on Inflammatory Signaling

During the development of hepatic inflammatory injury, several key factors, such as oxidative stress, osteopontin (OPN) and NF-κB signaling pathway, are known to devote the progression from steatohepatitis to fibrosis. The recruitment and activation of intrahepatic macrophages can generate more ROS, thus enhancing the hepatic levels of oxidative stress [104]. The high level of oxidative stress is known as a trigger for HSC activation [105]. Research has shown that OPN participated in activation of HSC and generation of extracellular matrix via multiple signaling pathways in steatohepatitis [106]. Activation of the NF-κB pathway can regulate the transcription of various pro-fibrogenic cytokines during inflammation, as well as interaction with TGF-β1 signaling, both of which can lead to hepatic fibrogenesis [107,108]. 

In several animal studies, TFA has been verified to simultaneously induce inflammatory damage in the liver and alteration of the abovementioned factors. A diet rich in TFA can provoke hepatic oxidative stress through an increase in ROS production, a decrease in antioxidant enzymes activity in NAFLD [109,110]. In our study, after treating with TFA-rich diet, the elevated expression of OPN was observed both in *Ppara*-deficient mice and HCVcpTg mice [20,111]. The other study has shown that feeding mice with a TFA-containing diet (maximum 13% energy intake from TFA) resulted in a notable infiltration of inflammatory cells in histological liver slices, increased hepatic expression of proinflammatory cytokines, and NF-κB activation [112]. After treating the murine model with liver-specific knockout of 11β-hydroxysteroid dehydrogenase type 1 gene with TFA-rich diet, increased expression of NF-κB and macrophage markers were detected [113]. 

In various isoforms of industrial TFA, trans-octadecenoic acids (c18:1 t) and alias elaidic acids (EAs) account for 80–90% of the total TFA content [114]. At the cellular level, EAs are added into culture medium to describe injurious effects in cells. Treated with 0.05 mM, EAs in human endothelial cells intensively activate NF-κB signaling [115]. The evidence from in vivo and in vitro studies has supported the credible mechanism that TFA initiated NF-κB signaling pathway in inflammation. Consequently, continuous consumption of TFA may cause persistent activation of NF-κB, subsequently resulting in steatohepatitis and steatofibrosis. 

### 4.2. The Impact of TFA on Lipid Metabolism

Studies have indicated that conspicuous collagen deposition around hepatocytes can be noticed in high-fat-diet-treated rat models. LDs accumulated in hepatocytes, while inflammatory cell infiltration was mild. The hepatic levels of α-SMA were positively correlated with LDs, suggesting that steatotic hepatocytes may directly activate HSC to promote liver fibrosis. Several studies have demonstrated that a TFA-enriched diet facilitated LD abundance in hepatocytes and led to hepatic steatosis. Enhanced expression of de novo FA synthesis-related genes and inhibited lipolysis-related genes expression in liver were positively associated with hepatic TG accumulation by TFA [111]. In addition, a TFA-rich diet can also stimulate cholesterogenesis via sterol regulatory element-binding protein 2 (SREBP-2), raise hepatic cholesterol concentrations, and accelerate NAFLD [116]. Treatment of human HepG2 cells with 0.1 mM EAs for 24 h exhibited a significant increase in SREBP-2 and hydroxy-β-methylglutaryl-CoA reductase [117], and treatment with 1.2 mM EAs after 12 h explored that EAs enhanced the catabolism of carbohydrate and promoted fatty acid biosynthesis [118]. As mentioned above, cholesterol overload in liver can lead to fibrosis, likely due to an aggravate inflammatory response, causing mitochondrial/ ER dysfunction and elevating oxidative stress and ER stress. Therefore, TFA may drive hepatic steatofibrosis partially by enhancing cholesterol synthesis. 

## 5. Future Perspectives

The prevalence of obesity and NAFLD/NASH is increasing worldwide, likely due to an increased consumption of fat-rich foods. Considering the complexity of lifestyle and several cofounding factors in human epidemiological studies, properly designed animal and cell experiments are mandatory to reveal the direct effect of different fat species and its underlying mechanisms. Although this review summarized the results of animal and cell studies demonstrating the influence of dietary cholesterol, SFA, and TFA on liver fibrogenesis and discussed the possible mechanisms, there are certain points to be clarified in future investigations. First, the impact of TFA on hepatocytes, KC, HSC, cholangiocytes, and LSEC should be clarified in more detail. We need to compare the actions to isolated cells between trans-form FA and cis-form in the identical carbon length. If trans-form FA is more toxic than cis-form, it is of interest to evaluate the mechanisms on how the structural changes in FA lead to cell toxicity. Second, the impact of dietary fat on crosstalk between hepatocytes and non-parenchymal cells and between liver and extrahepatic organs, i.e., gut-liver, mesentery (adipocyte)-liver, and spleen/pancreas-liver axes, should be addressed. The alterations in microbiota and microbiota-derived metabolites in response to dietary fat composition are also of interest. Lastly, future studies using animal models will enable us to establish appropriate dietary interventions. A dietary restriction of 30% downregulated inflammatory signaling, metabolic stress and senescence-associated secretory phenotype and suppressed steatosis-associated liver tumorigenesis in HCVcpTg mice [119]. The nutritional interventions targeting dietary fat may be useful to improve the outcome of NAFLD/NASH and other chronic liver diseases in humans. 

## 6. Conclusions

This review summarized possible mechanisms of three different lipids on parenchymal and non-parenchymal liver cells, which differentially contributed to the development of steatohepatitis and steatofibrosis. Significant differences among these three lipids were observed in the accumulation of lipids in the liver, contributing to the dysfunction of hepatocytes, M1 polarization of Kupffer cell, infiltration of inflammatory cell, activation of HSC, and the severity of hepatitis and fibrosis. There is no doubt that restriction of dietary cholesterol, SFA and TFA is beneficial to prevent steatohepatitis, fibrosis, and hepatocellular carcinoma. The further crosstalk between hepatology, lipidology, immunology, and nutritional research will pave a new avenue to confront liver disease. 

## Figures and Tables

**Figure 1 ijms-22-10303-f001:**
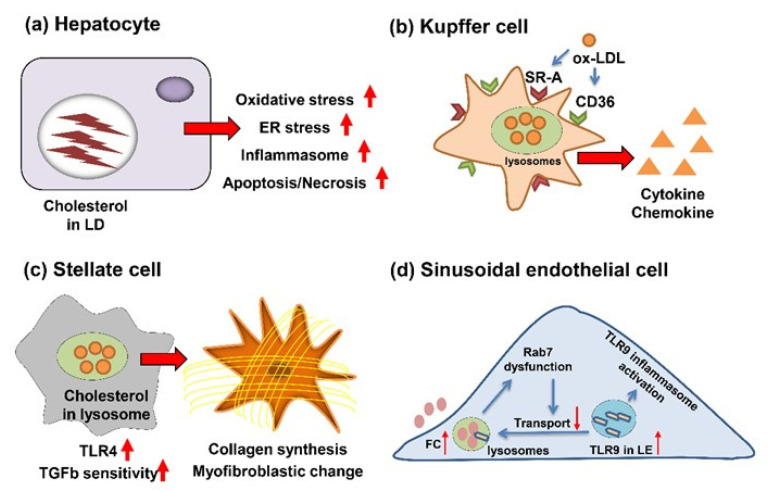
Impacts of dietary cholesterol on various cells in the liver. (**a**) In hepatocytes, accumulated free cholesterol enhances the progression of fibrosis through various damages. LD, lipid droplet; ER, endoplasmic reticulum. (**b**) In Kupffer cells, oxidized LDL is internalized by SR-A and CD36 and then pro-inflammatory and pro-fibrogenic cytokines and chemokines are secreted. (**c**) In hepatic stellate cells, intracellular cholesterol stimulates the stellate cell activation, mainly due to the increased TLR4 receptor levels and elevated sensitivity to TGFβ. (**d**) In hepatic sinusoidal endothelial cells, increased free cholesterol (FC) in lysosomes attenuates the degradation of TLR9 and then activates the TLR9/inflammation pathway. LE, late endosome. Arrows: ↑, Increased; ↓, Decreased.

**Figure 2 ijms-22-10303-f002:**
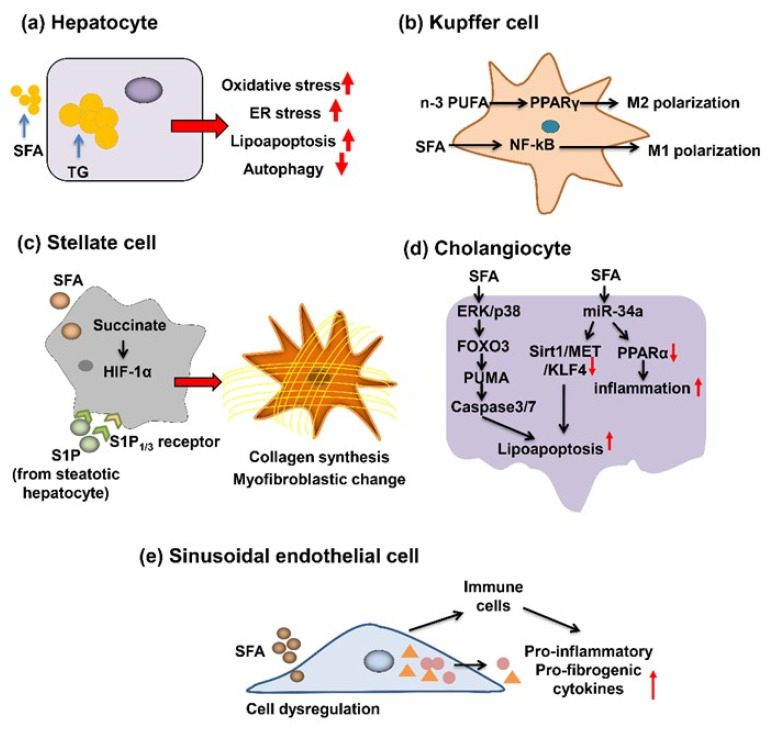
Impacts of dietary SFA on various cells in the liver. (**a**) In hepatocytes, SFA promotes fibrogenesis by enhanced oxidative/ER stress, lipoapoptosis and attenuated autophagy. (**b**) In Kupffer cells, n-3 PUFA polarizes to M2 in PPARγ-dependent manner, while SFA activates NF-κB to induce the pro-inflammatory M1 phenotype. (**c**) In hepatic stellate cells, palmitate and succinate can stimulate S1P/S1P receptors and HIF-1α axes and promotes myofibroblastic change. (**d**) In cholangiocytes, SFA promotes lipoapoptosis and inflammation. (**e**) In hepatic sinusoidal endothelial cells, SFA triggers dysregulation and changes into the pro-inflammatory and pro-fibrogenic phenotypes, partially through interaction with immune cells. Arrows: ↑, Increased; ↓, Decreased.

**Figure 3 ijms-22-10303-f003:**
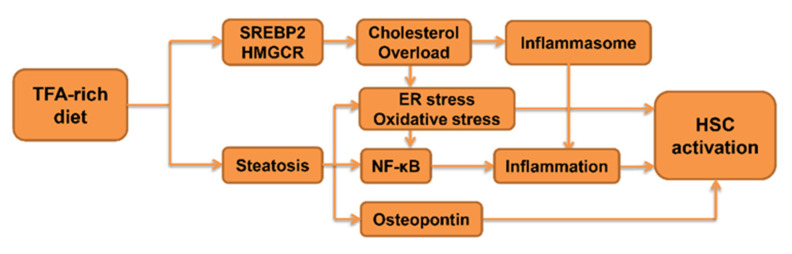
Proposed mechanism on how TFA promotes liver fibrogenesis.

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
