# Peer review of "Impact of Dietary Fat on the Progression of Liver Fibrosis: Lessons from Animal and Cell Studies"

_ijms, 2021, doi:10.3390/ijms221910303_

Round 1

Reviewer 1 Report

Thank you for your work.

Reviewer 2 Report

The authors have responded appropriately to the reviewer comments.

This manuscript is a resubmission of an earlier submission. The following is a list of the peer review reports and author responses from that submission.

Round 1

Reviewer 1 Report

The authors present a not-systematic review which accounts a group of studies regarding basic investigation in the field of liver fibrosis and the influence of diet. Although the work they have done is remarkable a few points should be clarified.

The first confusing term is, in my opinion, the title of the manuscript, since the review only includes animal and cellular studies and not human or epidemiological studies it could be changed to avoid confusion.

The authors should review some English expressions all along the manuscript and rethink the use of other like when they say that some animal model studies proved that that some dietary characteristic have some effect in liver fibrosis. Again, they are only talking about animal model based studies and they should clarify that the conclusions of these studies can only be hypothesis of the pathophysiology of liver fibrosis in humans.

Moreover, in some paragraphs the authors mixed cellular and animal model experiments, it could be of interest if they rewrite and divide in different paragraphs or sections to properly separate such different studies.

It is unclear what is the intention of each section of the manuscript. For example, it could be interesting if the authors could re-organize the sections talking about the main hypothesis of the role of cholesterol regarding each type of cell, then cite in proper order the main studies of each kind and finally conclude what did they think after their review about the role of each molecule in each cell type. The authors should remember that the conclusions should be carefully analyzed and they should take into account the huge limitations of this kind of studies.

In some sections, like in the paragraph entitled SFA´s action for KC, the authors include some sentences talking about potential therapeutic targets. This kind of argumentation must be rewritten in separate paragraphs and it could be also very interesting if the authors could write an entire section talking about potential therapeutic targets after their review.

Remember that every data should be supported with a proper reference. Some affirmations, like in line 160-161 of page 4 lack references to support them.

The authors include some information regarding gene expression regulation (like in line 234 page 7, where they talk about miR-34a) that are very confusing and makes very difficult the comprehension for the reader. If the authors want to analyze the effect of some dietary molecules over gen expression regulation pathways they should write a separate paragraph for this purpose.

Reviewer 2 Report

This is a brief but informative review of the impact of dietary fats on the different liver cell types and fibrosis. There are a couple of points that will strengthen the review, if addressed.

  1. The findings discussed are mostly from animal and cell studies. How do those findings correlate with the findings from human studies? It would be beneficial to discuss the knowledge in the context of the human condition.
  2. In the introduction, it is stated that NAFL is "benign and reversible". However, there is increasing evidence that NAFL may not be "benign", and may contribute to development of hepatocellular carcinoma without progression to NASH. This should be acknowledged and briefly described.
  3. Line 276: There is missing text, perhaps hidden by Figure 3?